# Habitual Choline Intakes across the Childbearing Years: A Review

**DOI:** 10.3390/nu13124390

**Published:** 2021-12-08

**Authors:** Emma Derbyshire, Rima Obeid, Christiane Schön

**Affiliations:** 1Nutritional Insight, Surrey KT17 2AA, UK; 2Department of Clinical Chemistry and Laboratory Medicine, University Hospital of the Saarland, D-66420 Homburg, Germany; rima.obeid@uks.eu; 3BioTeSys GmbH, Schelztorstrasse 54-56, D-73728 Esslingen, Germany; c.schoen@biotesys.de

**Keywords:** choline, essential nutrient, preconception, pregnancy, lactation, intake

## Abstract

Choline is an important nutrient during the first 1000 days post conception due to its roles in brain function. An increasing number of studies have measured choline intakes at the population level. We collated the evidence focusing on habitual choline intakes in the preconceptual, pregnancy, and lactation life stages. We conducted a review including studies published from 2004 to 2021. Twenty-six relevant publications were identified. After excluding studies with a high choline intake (>400 mg/day; two studies) or low choline intake (<200 mg/day; one study), average choline intake in the remaining 23 studies ranged from 233 mg/day to 383 mg/day, even with the inclusion of choline from supplements. Intakes were not higher in studies among pregnant and lactating women compared with studies in nonpregnant women. To conclude, during the childbearing years and across the globe, habitual intakes of choline from foods alone and foods and supplements combined appear to be consistently lower than the estimated adequate intakes for this target group. Urgent measures are needed to (1) improve the quality of choline data in global food composition databases, (2) encourage the reporting of choline intakes in dietary surveys, (3) raise awareness about the role(s) of choline in foetal–maternal health, and (4) consider formally advocating the use of choline supplements in women planning a pregnancy, pregnant, or lactating.

## 1. Introduction

Choline is an essential nutrient involved in critical physiological functions due to its role as methyl donor and precursor of acetylcholine and phospholipids [1]. Choline requirements in pregnancy and lactation are higher than in nonpregnant women due to the rapid division of foetal cells and active transport to the foetus and infant [2]. Growing evidence implies a role for sufficient maternal choline intake in foetal growth and development [3,4]. More recently, the American Medical Association and American Academy of Paediatrics have recognised that a failure to provide choline during the first 1000 days post conception could result in lifelong deficits in brain function regardless of subsequent nutrient repletion [5,6,7].

Focusing on brain development, a growing body of evidence suggests that choline is an important micronutrient and methyl donor that is required for normal brain growth and development, particularly during sensitive windows of life, including the childbearing years [1,8]. Choline has been acknowledged as being one of six key nutrients that are important for brain development, with nutrient effects appearing to be governed by the duration, timing, and severity of a deficiency or sufficiency [9]. A previous systematic review focusing on the first 1000 days of life concluded that choline could support normal brain development, help to protect against metabolic and neural insults such as alcohol exposure, and potentially facilitate neural and cognitive function [4]. Similarly, other authors recognised that large proportions of choline-derived phospholipids, e.g., phosphatidylcholine and sphingomyelin, are required for cell division, growth, and myelination, which occur rapidly during foetal development [3]. The foetal origins of memory hypothesis proposes that maternal and infant dietary intakes of choline influence brain development which could permanently modulate brain function of the offspring, i.e., resulting in cognitive and memory deficits with mechanisms likely to involve DNA methylation and alterations to gene expression, as well as stem-cell proliferation and differentiation [10]. Underpinning studies in rodents further demonstrate that a higher choline intake during pregnancy facilitates cognitive function and offsets memory decline with advancing age [11].

In 2016, the European Food Safety Authority (EFSA) set an adequate intake (AI) of 400 mg/day for all adults, 480 mg/day for pregnant women, and 520 mg/day for lactating women [12]. The AIs for choline according to the United States Institute of Medicine (IOM) for nonpregnant, pregnant, and lactating women are 425 mg/day, 450 mg/day, and 550 mg/day, respectively [13]. The primary criterion for setting the AI for choline was the prevention of liver damage as assessed by measuring serum levels of liver enzymes, whilst a higher AI during pregnancy and lactation was justified by the extra requirements of the foetus and infant [13]. Due to inadequate data on choline intake in the population, it was not possible to derive an estimated average requirement (EAR) and recommended dietary allowance (RDA) for choline. An RDA for choline would be expected to meet the needs of 97–98% of individuals in a life stage and gender group. Since 2004, data on choline intake have become available from several populations.

Choline appears to be an under-consumed and overlooked nutrient [14,15] not only in Americans [16,17,18] and Australasian populations [19,20], but also in Europeans [21]. In 2018, Wiedeman et al. published a review on dietary choline intake across the lifecycle which included adults and other age groups (toddlers 1–3 years, children 4–9 years, adolescents 10–18 years, pregnancy, lactation, and the elderly >65 years) [22].

The present publication collates habitual choline intake data specific to the childbearing years using data from human studies published between 2004 and 2021.

## 2. Materials and Methods

### 2.1. Search Strategy and Databases

The search was conducted in the National Library of Medicine National Centre for Biotechnology Information (PubMed.gov) platform on the 8th September 2021 and updated on the 15th of November 2021 using the advanced search builder and the terms shown in Appendix A. We further searched reference lists of previous reviews and relevant studies.

The publication inclusion criteria were human studies in English language reporting habitual choline intakes from the diet in women aged 16–50 years (as a main aim) and published between 2004 (1 January 2004) and 2021 (15 November 2021). This timeframe was specified due to the United States Department of Agriculture (USDA) Food Composition Databank being completed in 2004 [23]. The population(s) of focus were women of childbearing age (16 to 50 years) defined as those who were nonpregnant, pregnant, or lactating [24].

We planned to include studies published in English language without restriction on the country of origin. We included all study types (observational and interventional) that reported mean habitual choline intakes during the specified period. We included studies where mean habitual choline intakes were recorded without supplements (from foods only; FO) and when a combination of habitual intakes and multivitamin or mineral supplements was used (from food and supplements combined; F+S).

We excluded animal studies, cell culture studies, narrative reviews, case reports, and case series publications. We also omitted studies unrelated to the target population of the present review, e.g., cardiovascular or fatty liver disease. Studies with an excessively high (>400 mg/day choline) or low (200 mg/day choline) intake from dietary sources alone were excluded due to intakes likely bring skewed by cultural factors or methodological limitations, e.g., small sample size or accuracy of dietary assessments. This included, for example, research where participants frequently ingested “menudo” (a Mexican broth using beef stomach) [25]. Studies were also excluded where the mean intake was not reported or could not be derived.

E.D. screened the titles and abstracts for relevance. The full texts of relevant articles were then reviewed, and data were extracted by E.D. and verified by R.O.

### 2.2. Data Extraction

A standardised spreadsheet was used to extract the following data: PubMed Identifier, year, first author, title, country of the participants, time period of intake assessment or pregnancy trimester, mean age, study design, sample size, how choline intake was assessed (i.e., food frequency questionnaires (FFQ), semi-quantitative food frequency questionnaire (SQ FFQ), dietary recall), and sources of choline (i.e., foods only or from foods and supplements). In addition, means and standard deviations (SDs) or medians and interquartile ranges (IQRs) of daily choline intakes (mg/day) were extracted. These were then compared against EFSA and IOM AIs for three target groups: nonpregnant, pregnant, and lactating women.

## 3. Results

### 3.1. Search Results

The search for publications was undertaken on 8 September and updated on 15 November 2021. Using the specified search terms, 91 publications were yielded. A further nine relevant studies were identified through screening reference lists. As shown in Figure 1, after screening, 74 publications were excluded, leaving 26 publications for inclusion within the main review (Table 1, Table 2 and Table 3).

### 3.2. Childbearing Age

Eleven studies measured choline intakes amongst women of childbearing age [17,19,21,25,26,27,28,29,31,32,33]. Seven studies recorded choline intakes derived from food sources, [19,21,25,28,31,32,33]. Highest median choline intake (760 mg/day in cases and 722 mg/day in controls) was reported in women from the Texas–Mexico border, possibly due to menudo (a traditional Mexican broth made with beef stomach) [25]. After excluding this study, choline intakes in the remaining seven publications ranged from 244 to 443 mg/day from food sources [19,21,28,31,32,33].

The remaining four publications derived choline intakes from food and supplemental sources [17,26,27,29]. When supplements were included, choline intakes ranged from 250 mg/day in research conducted by Wallace et al. (2016) [17] to 308 mg/day in the cross-sectional study undertaken by Lee et al. (2010) [29].

Focusing on European data, Vennemann et al. (2015) [21] estimated choline intake from food sources in females (age 18–65 years) from Finland, France, Ireland, Italy, Netherlands, Sweden, and the United Kingdom. Mean choline intakes from food sources in this European-wide population study ranged from 291 mg/day in France to 374 mg/day in Sweden. Similar choline intake estimates were reported in women from countries outside the European region. American studies amongst women of childbearing age have reported choline intakes ranging between 250 and 443 mg/day [17,26,29,31,32,33].

In Australia, mean choline intakes for women of childbearing age (16–44 years) were 244 mg/day (95% CI 246–255) from food sources [19]. Only 4.75% women achieved choline intakes equivalent to or higher than the country’s AI of 425 mg/day [19].

### 3.3. Pregnancy

Sixteen studies recorded choline intakes during pregnancy [16,18,19,20,34,35,36,37,38,39,40,41,42,43,44,45]. Seven documented choline intakes in the first trimester [35,37,38,39,41,43,45], along with five in the second trimester [37,38,41,42,43], five in the third trimester [37,38,39,40,41], and six in pregnancy overall [16,18,19,20,34,44]. Across the trimesters, changes in choline intake were generally very small. Two studies reported a 12–13 mg/day increase in choline intake between the first and thirst trimesters [37,41]. Two publications recorded a decline in mean choline intakes between the first and third trimesters (−4 and −12 mg/day) [38,39].

Mean choline intake estimates for pregnant adults from the United States ranged from 281 mg/day to 332 mg/day [16,18,34,38,43]. Intakes were similar in pregnant Canadians with mean intake estimates ranging between 306 mg/day and 383 mg/day [39,41,42,46]. Moore et al. (2020) [35] reported a median choline intake (from foods and supplements) of 338 mg/day in UK pregnant women. Median dietary choline intake in Australian pregnant females (aged 19 to 50 years) was 251 mg/day [19]. Similar intakes were reported in human studies conducted in Belgium (mean intakes: 274 mg/day (first trimester)–280 mg/day (third trimester)) [37], China (median 255 mg/day) [36], Jamaica (mean 279 mg/day) [45], and marginally higher in New Zealand (median 310 mg/day) [20].

Mean choline intake in Latvian pregnant adolescents was 336 mg/day [44]. Lowest choline intakes were reported by young women (21–25 years) in their third trimester in Bangladesh (190 mg/day) [40].

The inclusion of dietary supplements did not substantially increase choline intakes. When derived from foods only, mean choline intake estimates for pregnant adults ranged from 190–383 mg/day [19,20,34,36,38,40,42]. In studies when estimates included choline from foods and supplements, mean intakes ranged from 268 to 353 mg/day and, thus, did not change substantially [16,18,35,37,39,41,43,46].

### 3.4. Lactation

Three studies recorded the mothers mean choline intakes during lactation [19,37,41]. Postpartum intakes of choline were recorded in research conducted in Belgium [37], Canada [41], and Australia [19]. Lowest intakes were reported in Australia (median 257 mg/day) [19], and highest intakes were reported in Canada (mean 346 mg/day) [41].

## 4. Discussion

Choline is a critical nutrient during pregnancy and lactation because it plays a central role in foetal and child development [1,3,4,8]. Therefore, we investigated the current evidence on whether women of childbearing age, pregnant women, or lactating women are achieving sufficient choline through their diet or a combination of diet and supplements. The majority of studies we identified among these groups of women (24 studies) reported average choline intakes below the AIs as recommended by the EFSA and IOM (Appendix A). Two studies reported mean choline intake estimates over 400 mg/day [25,32]. These were conducted in Mexican American populations or the United States. Lavery et al. (2014) reported the highest average choline intakes (818 mg/day in controls), which was attributed to the consumption of “menudo”—a traditional Mexican soup made into a broth using beef stomach [25]. The remaining study by Fischer et al. (2005) reported a mean intake of 443 mg/day in nonpregnant women (*n* = 16), only just exceeding the AI for nonpregnant women (400 mg/day) but below the EFSA AI for pregnancy (i.e., 480 mg/day) [32,33]. With the exception of these two studies, choline intakes from studies in the US were similar to those reported in European studies. Five percent of the women had choline intake above 631 mg/day in Sweden, above 578 mg/day in Finland, and above 543 mg/day in the Netherlands [21].

Lowest intakes were reported during pregnancy in Bangladesh (190 mg/day) by Goon and Dey (2014), who described that the retrospective 24 h dietary recall may have been subject to limitations [40]. The average intakes from the remaining studies ranged from 233 mg/day to 383 mg/day. The reported studies show large variations in average choline intakes between different countries and studies (or ethnic groups) within one country. Moreover, European studies suggest the presence of a south to north gradient, with intakes likely to be slightly higher in Nordic countries than in southern Europe. Choline intake recommendations on a population level rely mainly on intake data from representative studies, especially as there are no optimal biomarkers to measure choline status at present.

Overarching results clearly demonstrate that most women of childbearing age across the globe are not likely to meet the current AIs for choline. Moreover, the studies have shown that average choline intakes are not higher in pregnant and lactating women compared to nonpregnant women, suggesting the urgent need to stress the recommendations of achieving sufficient choline intake in women. Higher maternal choline intake has been associated with a lower odds ratio of neural tube defects [33], and experimental choline deficiency is associated with brain malformation [47,48]; thus, it can be argued that choline should be added to prenatal supplements.

The gap in choline intake is approximately 70–100 mg/day, which needs to be replaced via prenatal supplements. There are several supplemental forms of choline such as choline bitartrate, choline chloride, and phosphatidylcholine. There is currently no sufficient evidence that either one of these forms is preferable. However, the disadvantage of phosphatidylcholine is that a large dose is required to achieve the required choline intake, which could be a limitation when combining phosphatidylcholine with other vitamins, as two capsules may need to be supplemented.

In terms of dietary trends, whilst plant source foods can provide some choline [3], it is well recognised that animal-derived foods such as meat, eggs, milk, and fish tend to provide more choline per unit weight than plant-based foods such as fruit, vegetables, and grains [30,49,50,51]. The global trend to reduce animal-source foods implies that many women of childbearing age could have further decrements in choline intakes [14,15]. For example, Lecorguille et al. (2020) observed that the “vegetarian tendency” dietary pattern was associated with lower intake coefficients for choline, vitamin B12, and methionine [52]. Metabolomic research showed that an omnivorous breakfast resulted in higher choline concentrations in serum compared with lacto-/ovo-vegetarian and vegan options [53]. A cross-sectional evaluation of 74 US lactating women showed large between-individual variations in total water-soluble choline levels in samples of breast milk (range between 4 and 301 mg/L). While total water-soluble choline forms in breast milk did not significantly differ with maternal diet, mean choline derived from glycerophosphocholine was 20% higher, and mean choline from phosphocholine was 12% lower in vegan mothers compared to nonvegetarian mothers [54]. Another study showed that concentrations of water-soluble choline forms in mature milk did not significantly differ between lactating women in Canada and Cambodia despite likely lower choline intakes in Cambodian women compared to Canadian women [55]. Altogether, these studies are difficult to compare due to variations in intakes and sources of choline in maternal diet, as well as due to analytical methods of breastmilk choline without taking fat-soluble choline derivatives into account.

Findings show that it is necessary to increase knowledge about choline during these important life stages. Whilst diets may improve somewhat in pregnancy, dietary planning tends to become more challenging postpartum [56,57]. Swedish research showed that women’s diet quality declined postpartum, mainly due to increased intakes of discretionary foods [56]. Similarly, amongst Australian first-time mothers, only 8.6% met guidelines for combined fruit and vegetable intake, indicating a gap in the distribution of healthy eating advice after birth [57]. Subsequently, particular attention should be paid to these life stages. These findings imply that women are not fully adopting basic healthy eating guidelines; thus, much will need to be done to raise awareness about the role(s) of specific nutrients such as choline.

The present study identified gaps in knowledge, sources of heterogeneity, and possible topics to be improved in future studies. The limitations of the present study are its inclusion of only English literature, the search being limited to the PubMed platform and reference searches, and the lack of a plan to perform a systematic search combined with quantitative data analysis.

### Limitations in the Evidence and Gaps in Knowledge Surrounding Choline Intake

Most studies reported that they used ‘validated’ methods to determine choline intakes. Two publications, for example, used the Willett SQ FFQ [26,28,58]. However, it was unclear whether these were fully validated for their accuracy in relation to obtaining suitable energy intakes and/or preventing under-reporting. In Cape Town, one quantitative FFQ appears to have been developed and validated to determine choline intakes in pregnant women [59]. This specific tool comprised 10 food groups (beef, lamb, chicken, processed meat, fish, eggs, vegetables, fruit, dairy, and other items) and demonstrated reliability and validity, thus having potential to be used in other communities where choline inadequacy may exist [59]. Weighed-intake food records could also be used as these tend to provide precise estimates of portion consumed, with evolving technologies helping to ease recording burdens placed on participants [60]. Levels of under- and over-reporting also need to be accounted for, using methods such as the Goldberg equation [61]. Plasma free choline can also be used as a biomarker to assess choline status but can be highly variable [62]. The majority of studies used the USDA database to assess data on the choline composition of foods or software programs into which this information was integrated. Several publications also used data from Zeisel et al. (2003), who published values in relation to the concentrations of choline compounds found in foods [30]. It is worth noting that the USDA database does not yet contain food composition values for all foods. Given the changing dynamics to current food markets (moving away from animal-derived towards plant-derived proteins), there is a need to urgently add to and update such databases.

Updated studies are needed using rigorous methodologies to estimate the intake of choline and choline-containing phospholipids from different foods and link the intake and bioavailability to the requirements. Future studies also need to reduce variability in methods used to estimate choline intake and establish country-specific food tables that contain choline and choline-related compounds. Cooking methods have been found to reduce the relative percentage from free choline whilst increasing the contribution of phosphatidylcholine to total choline for most pulses [49]. Cooking may cause up to ±30% variability of choline content per serving compared with US Department of Agriculture database figures [49]. Mincing raw vegetables can also reduce phosphatidylcholine content by activating phospholipase D, thus releasing free choline and phosphatidic acid [30]. Moreover, variations in gut microbiome, polymorphisms, and other metabolic factors could influence the bioavailability and turnover of choline in the body [63,64]. In addition, lifestyle factors such as adherence to a vegan/vegetarian diet are associated with consuming different sources and forms of dietary choline.

In addition, several studies noted that supplements were included but did not provide a breakdown of intake data from dietary sources per se and dietary sources and supplements separately. When supplements were used, studies rarely specified ‘how much’ choline was provided from supplements. As separate choline supplements are not yet widely available or known about, and as choline is not yet presently added to most multivitamin and mineral preparations, this was potentially overlooked [5,14]. However, it would be useful to segregate and account for this in future studies.

On a final point, the reference intake of choline increases during pregnancy and lactation compared to that in nonpregnant women. An RDA for choline has not yet been established due to a lack of population-representative intake data. Thus, population intake data, especially in vulnerable groups may help in defining an RDA for choline. Our study mapping studies on choline intake throughout the childbearing age is a first step towards establishment of average population intakes of choline.

## 5. Final Evaluation and Conclusions

In summary, average choline intakes in women from different countries across the globe appeared to be lower than IOM or EFSA guidance, and intake values were generally not higher in studies among pregnant and lactating women compared to studies among nonpregnant women, thus indicating that a considerable proportion of women are not meeting the AI. This is concerning given the growing body of evidence relating choline to foetal and child development [1,4,5,8,65]. Taken collectively, a number of urgent measures are needed. This includes (1) adding and improving data quality in food composition tables to facilitate the collection of information on choline intake from different populations, (2) integrating and reporting choline intakes in dietary survey across the globe, (3) updating health policies and embedding choline within these to raise awareness about its role(s) in foetal–maternal health, and (4) formally recommending that women planning a pregnancy, as well as pregnant or lactating women, should supplement their diets with choline.

## Figures and Tables

**Figure 1 nutrients-13-04390-f001:**
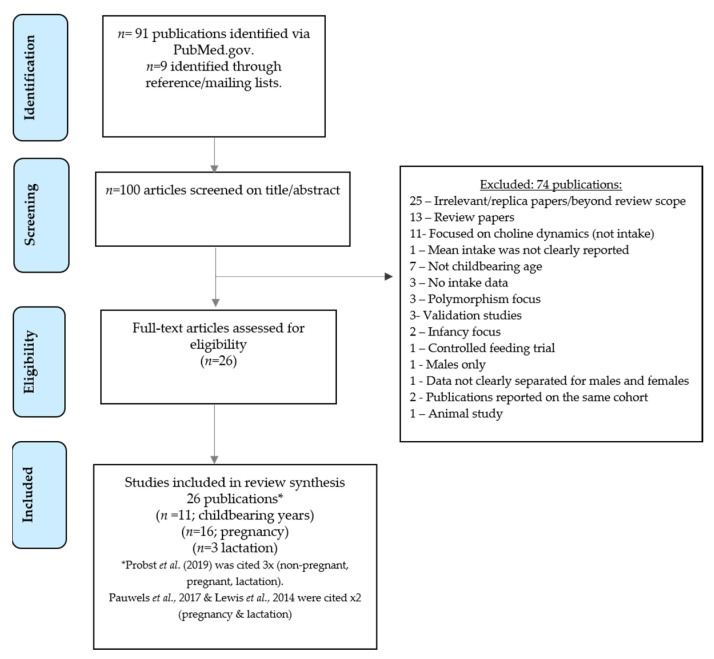
Study flow diagram.

**Table 1 nutrients-13-04390-t001:** Habitual choline intakes in nonpregnant women of childbearing age.

First Author, Country	Age, Years	Study Design, Cohort Name	Sample Size	Method and Duration/Time Period	Database Used to Estimate the Intake	Source(s) of Choline	Choline Intake (mg/day)
Petersen et al. (2019), United States [26]	Range <25 years (12.4%) 25–34 years (68.3%) ≥35 years (19.4%)	Case–control study (the Slone Birth Defect study 1975–2014)	Mothers of *n* = 164 NTD cases (live births, foetal deaths, elective terminations), and *n* = 2831 control infants	SQ FFQ (the Willett FFQ) to capture the intake in the past 6 months	Harvard T.H. Chan School of Public Health Nutrition Department’s Food Composition Table.	F+S	Mean (SD) = Controls: 275 (68) Cases: 273 (67) Choline intake data were based on women with folic acid intake >400 µg/day
Probst et al. (2019), Australia [19]	Range 16–44 years	Cross-sectional study based on The Australian Health Survey 2011–13	*n* = 2210	2 day intake data	Sourced studies and global food composition databases and compared these with data from Australian foods (from AUSNUT 2011–13 Australian Food)	FO	Median (IQR) = 233 (103)
Oyen et al. (2017), Norway [27]	Range 46–49 years	Cross-sectional study based on The Hordaland Health study	*n* = 1600	FFQ (habitual diet from past year) validated: 169 items; frequency of consumption specified per day, week, or month	Choline intakes estimated according to the 2008 USDA choline database	F+S	Median (IQR) = 255 (63)
Gao et al. (2016), Canada [28]	Mean = 43.7 years	Cross-sectional CODING study (Complex Diseases in the Newfoundland population: Environment and Genetics) study	*n* = 2232–2295 (sample size range)	SQ FFQ (the Willett FFQ) 124 items; data from past 12 months	NutriBase Clinical Nutrition Manager	FO	Mean (SD) = 292 (213)
Wallace et al. (2016), United States [17]	Range 19–30 years Range 31–50 years	Cross-sectional study based on the National Health and Nutrition Examination Survey (2009–2012)	*n* = 1096 (19–30 years) *n* = 1794 (31–50 years)	Two 24 h dietary recalls with trained interviewers	Various USDA food composition databases	F+S	Mean (SD) = 19–30 years: 250 (166); 31–50 years: 278 (169)
Vennemann et al. (2015), Europe [21]	Range 18–≤65 years	Cross-sectional dietary surveys in each country	Finland *n* = 710; France *n* = 1340; Ireland *n* = 640; Italy *n* = 1245; Netherlands *n* = 1034; Sweden *n* = 807; UK *n* = 706	3 day dietary record 7 day dietary record 4 day dietary record 3 day dietary record 24 h dietary recall 4 day web record 4 day dietary record	USDA Database (2013)	FO	Mean (5th, 95th percentiles) = Finland: 344 (177, 578); France: 291 (162, 440); Ireland: 318 (166, 485); Italy: 293 (153, 463); Netherlands: 334 (185, 543); Sweden: 374 (186, 631); UK: 294 (145, 478)
Lavery et al. (2014), Mexican Americans [25]	Range <20 years (24.7%) 20–24 years (34.2%) 25–29 years (24%) >30 years (35%)	Case–control study	Mothers of *n* = 184 cases with NTDs and *n* = 225 controls	FFQ: 98 items ascertaining food frequency for 3 months before their conception date to 3 months postpartum	USDA database containing 630 food items and six choline metabolites.	FO	Median (IQR) = Controls: 760 (456); Cases: 722 (689)
Lee et al. (2010), United States [29]	Range 29–86 years	Cross=sectional study in the sixth examination (1995–1998) of the Framingham Offspring Study	*n* = 1407	FFQ, validated: 130 items; frequency of consumption over the past year	Choline composition data values published by Zeisel et al. [30] and from the USDA 2008 Database	F+S	Energy adjusted mean (SD) = 308 (56)
Chiuve et al. (2007), United States [31]	Range 30–55 years	Cross-sectional study based on the Nurses’ Health Study	*n* = 1477	SQ FFQ, validated: undertaken every 4-years	Choline composition data values published by Zeisel et al. [30] and from the USDA 2008 Database	FO	Energy adjusted median (range of 3rd quintile) = 323 (311–334)
Fischer et al. (2005), United States [32]	Range 18–67 years	Ad libitum dietary intake in the study centre	*n* = 16	Choline content was measured in the foods and compared to estimates from a pre-study 3 day food record	The Food Processor SQL program using USDA Nutrient Database and Zeisel et al. [30] data	FO	Mean (SD) = 443 (88)
Shaw et al. (2004), United States [33]	Women recruited after delivery, age not reported	Case–control study	Mothers of *n* = 424 cases with NTDs, mothers of *n* = 440 controls	FFQ: 100 items used to assess frequency and portion size consumed 3 months before conception	Used choline values published by Zeisel et al. [30]	FO	Mean (SD) = Controls: 409 (179); Cases: 377 (176)

FFQ, food frequency questionnaire; FO, foods only, F+S, foods and supplements; IQR; interquartile range; NTD, neural tube defects; SD, standard deviation; SQ FFQ, semi-quantitative food frequency questionnaire; USDA, United States Department of Agriculture.

**Table 2 nutrients-13-04390-t002:** Habitual choline intakes in pregnant women.

First Author, Country	Age, Years	Study Design	Sample Size	Method and Duration/ Time Period	Database Used to Estimate the Intake	Source(s) of Choline	Choline Intake (mg/day)
Fawcet et al. (2020), United States [34]	Not reported.	Prospective longitudinal study	*n* = 251	3 day food records completed during the 1st, 2nd, and 3rd trimester	Nutrient Data System for Research	FO	Mean (SD not reported) = 281
Moore et al. (2020), United Kingdom [35]	Mean (SD) = 31.4 (4) years	Be Healthy in Pregnancy (B-HIP) study, baseline data of a RCT	*n* = 232	3 day weighed diet record (2 week days and 1 weekend day) completed at 12–17 weeks gestation	Nutritionist Pro™ diet analysis software	F+S	Median (min, max) = 338 (120, 1016)
Zhu et al. (2020), China [36]	Mean (SD) = 28 (4) years	Case–control study (SQ FFQ retrospective collecting dietary intake during pregnancy)	Mothers of *n* = 157 term controls	SQ FFQ, validated, 120 food items including the most common foods in the Chinese diet conducted no later than 3 days after parturition	China Food Composition and the USDA Food Composition Database	FO	Energy-adjusted choline intake in the controls Median (IQR) = 255 (70)
Probst et al. (2019), Australia [19]	Range 19–50 years	Cross-sectional study based on data from the Australian National Nutrition and Physical Activity Survey 2011–12	*n* = 116	2 day intake data (data filtered for women who were pregnant at the time of the survey)	Sourced studies and global food composition databases, compared with data for Australian foods to create a choline database	FO	Median (IQR) = 251 (111)
Bailey et al. (2019), United States [16]	Range 20–40 years	Cross-sectional study based on data from NHANES (2001–2014)	*n* = 533	24 h dietary recall ×2 (taking part in the What We Eat in America survey)	USDA Food Composition Database 2019	FO	Mean (SD) = 321 (231)
Pauwels et al. (2017), Belgium [37]	Range 25–41 years	A longitudinal Maternal Nutrition and Offspring’s Epigenome study (MANOE)	1st trimester: *n* = 94 2nd trimester: *n* = 85 3rd trimester: *n* = 82	FFQ, validated: 51 items completed at 11–13, 18–22, and 30–34 weeks of pregnancy	Not reported	F+S	Mean (SD) = 1st trimester: 274 (72); 2nd trimester: 268 (68); 3rd trimester: 280 (78)
Wallace et al. (2017), United States [18]	Range 13–44 years	Cross-sectional study based on data from the 2009–2014 and 2005–2014 NHANES (2009–2014 and 2005–2014) datasets	*n* = 593	24 h dietary recalls ×2	Various USDA food composition databases used	F+S	Mean (SD) = 319 (241)
Groth et al. (2017), United States [38]	Range 18–36 years	Prospective observational study, secondary analysis of the Limiting the Phenotypic Effect of Pregnancy-Related Weight Gain	1st trimester: *n* = 90 2nd trimester: *n* = 68 3rd trimester: *n* = 67	24 h dietary recalls ×3 at three timepoints: early (<22 weeks), mid (24–29 weeks), and late (32–37 weeks) pregnancy.	Nutrition Data System for Research software 2009	FO	Mean (SD) = 1s^t^ trimester: 318 (68); 2nd trimester: 289 (28); 3rd trimester: 306 (28)
Masih et al. (2015), Canada [39]	Mean (SD) = 32 (5) years	Prospective observational study, the Prenatal Folic Acid Exposure on DNA Methylation in the newborn infant study	*n* = 290	SQ FFQ, validated: 110 items; recall of habitual intakes between 0–16 and 23–27 weeks gestation	Nutrition Quest used the nutrient composition data primarily from version 1.0 of the USDA Food and Nutrient Database	F+S	Mean (SD) = 1st trimester: 306 (127); 3rd trimester: 302 (122)
Goon et al. (2014), Bangladesh [40]	Range 21–25 years	Cross-sectional study	*n* = 103	24 h dietary recall in the 7th, 8th, or 9th months of pregnancy	USDA Food Composition database	FO	Mean (SD) = 190 (98)
Lewis et al. (2014), Canada [41]	Range 17–30 years (45.8%) 31–45 years (54.2%)	Prospective cohort study, the Alberta Pregnancy Outcomes and Nutrition (APrON) cohort study	1st trimester: *n* = 123; 2nd trimester: *n* = 562; 3rd trimester: *n* = 493	24 h dietary recall using the multiple-pass method in each trimester	The Alberta database used the USDA Database for the Choline Content of Common Foods, Release 2 (634 foods)	F+S	Mean (SD) = 1st trimester: 340 (148); 2nd trimester: 349 (154); 3rd trimester: 353 (144)
Mygind et al. (2013), New Zealand [20]	Range 18–40 years	Baseline part of dietary data collection for a folate intervention study	*n* = 125	3 day weighed food record Two weekdays and one weekend day	USDA Choline Content of Common Foods, Release 2, 2008	FO	Median (IQR) = 310 (87)
Wu et al. (2012), Canada [42]	Not reported	Prospective study	*n* = 154	FFQ, women enrolled from 16 weeks gestation	USDA Choline Content of Common Foods, Release 2, 2008	FO	Mean (SD) = 383 (99)
Villamor et al. (2012), United States [43]	Mean (SD) = 33 (5) years	Project Viva longitudinal study	1st trimester: *n* = 1148 2nd trimester: *n* = 1083	FFQ in the first and second trimester (26–28 weeks gestation)	Harvard nutrient composition database	F+S	Mean (SD) = 1st trimester: 332 (63); 2nd trimester: 325 (64)
EFSA (2016), Latvia [12,44]	Range 15–45 years	Food Consumption data from the EFSA European Comprehensive Food Consumption Database	*n* = 990	24 h dietary recall in pregnancy	Not reported	FO	Pregnant adolescents: Mean = 336; Pregnant women: Mean (5th, 95th percentiles) = 356 (200, 592)
Gossell-Williams et al. (2005), Jamaica [45]	Range 18–32 years	Observational study	*n* = 16	FFQ (Jamaican foods) recruited at 10–15 weeks gestation	USDA Food Composition database	FO	Mean (SD) = 279 (116)

EFSA, European Food Safety Authority; FFQ, food frequency questionnaire; FO, foods only; F+S, foods and supplements; IQR; interquartile range; NHANES, National Health and Nutrition Examination Survey; RCT, randomised controlled trial; SD, standard deviation; SQ FFQ, semi-quantitative food frequency questionnaire; USDA, United States Department of Agriculture.

**Table 3 nutrients-13-04390-t003:** Habitual choline intakes in lactating women.

First Author, Country	Age, Years	Study Design	Sample Size	Method and Duration/Time Period	Database Used to Estimate the Intake	Source(s) of Choline	Choline Intake (mg/day)
Probst et al. (2019), Australia [19]	Range 19–50 years	Cross-sectional study based on data from the Australian National Nutrition and Physical Activity Survey 2011–12	*n* = 110	2 day intake data analysed from ‘lactating mothers’	Sourced studies and global food composition databases, compared these with data for Australian foods to create a choline database	FO	Median (IQR) = 257 (100)
Pauwels et al. (2017), Belgium [37]	Range 25–41 years	The Maternal and Offspring’s Epigenome prospective, observational cohort study	6–8 weeks PP: *n* = 79; 6 months PP: *n* = 60	FFQ, validated 51-items conducted at 6–8 weeks and 6 months postpartum	Not reported	F+S	Mean (SD) = 6–8 weeks PP: 278 (76); 6 months PP: 268 (60)
Lewis et al. (2014), Canada [41]	Range 17–30 years (45.8%) 31–45 years (54.2%)	Alberta Pregnancy Outcomes and Nutrition (APrON) cohort study	*n* = 488	24 h dietary recall using the multiple-pass method	The Alberta database was developed using the USDA Database for the Choline Content of Common Foods, Release 2 (634 foods)	F+S	Mean (SD) = 346 (151)

FFQ, food frequency questionnaire; FO, foods only, F+S, foods and supplements; IQR; interquartile range; PP, postpartum; SD, standard deviation; USDA, United States Department of Agriculture.

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
