# Peer review of "Habitual Choline Intakes across the Childbearing Years: A Review"

_nutrients, 2021, doi:10.3390/nu13124390_

Round 1

Reviewer 1 Report

The authors reported a scoping review of habitual choline intakes of women of reproductive age based on life stages – nonpregnant & nonlactating women of reproductive age, pregnant women, and lactating women. The methodology was straightforward with clearly defined scope. The review provides a nice summary of literature on this topic between 1997 and Sep 2021 without much additional synthesis. It would be further strengthened by addressing the below comments.

  • In many cases, dietary intake data are not normally distributed. Suggest extracting and report median, IQR, minimal and maximum along with mean and SD/SE, if those data are reported.
  • Please include the database for the choline content of foods used in calculating choline intake for each study. Did all studies used USDA Database for the Choline Content of Common Foods, Release Two? If not, please comment on the potential bias introduced.
  • Table 2
    • Clearly define the abbreviation of CB.
    • Under the “Study Design” column, it’s fine to include the study name, but please specify the study design. For example, what is the study design for “Hordaland Health Study” “Ad libitum study”? For cohorts, it would be good to specify if these are prospective or retrospective cohorts. Please go through the table and update accordingly.
  • Lines 107 - 108: Appears to miss a word “used”. Carmichael et al (2010) used a shorter xxx.
  • Lines 142-144 and table. For the ones included choline intake from supplements. Please provide additional information on supplements as choline-containing supplements are a recent phenomenon. It’s not clear if the differences observed as stated in lines 142-144 is due to the true choline-containing supplement use or due to different methodology was applied in calculating choline intakes.
  • Provide discussions and comments on different methodologies in estimating habitual choline intake and how this could introduce biases. It would be good to synthesize and provide recommendations on the most appropriate tools to use to estimate choline intake (e.g., FFQ vs. recall or food diary).
  • Please add “strength and limitations” to the Discussion session.
  • How does choline intake correlate with choline status? The discussion could be strengthen by covering this topic.

Author Response

Thank you.

A point by point descripta has been provided in the attached word document.

Reviewer 2 Report

This scoping review is focused on choline intake in childbearing age women. A previous choline intake review, which included this age group, was published in 2018 and the present manuscript includes a number of more recent studies. I have several questions and suggestions to improve this manuscript.

Introduction

Short and concise introduction. However, some points should be explained and discussed in greater depth later in the manuscript.

Materials and Methods

Please make sure that you are familiar with the recommendations and terminology used in a scoping review. Pubmed is referred to as a database, however, is a platform (similar to Ovid). Some of the recommendations for scoping reviews include developing a protocol, using at least 2 databases, and publications to be screened/reviewed for more than 1 author.

The rationale provided for the time frame selected is inappropriate, as the first available database for choline content in foods was in 2004 and updated in 2008. Therefore, it would make sense that the time frame will start in 2004 and not in 1997.

Selecting studies in the English language and then mentioning we planned to include studies from all countries ... is contradictory.

As mentioned that choline intake will be reported from foods only and plus supplements, I expected a greater detail level in the results and discussion.

Only mentions that choline intakes will be compared to EFSA. How about IOM?

Results

Figure 1 is mentioned in the text, but the figure legend is missing.

To improve the readability, I would recommend dividing Table 2 into each category (not pregnant/lactating, pregnancy, lactation).

The quality and level of detail provided should be improved.

Table 2 needs to include footnotes, including all the abbreviations used. In addition, the use of abbreviations should be consistent in the whole manuscript, including the table. For example, it is used semiquantitative FFQ and SQ FFQ.

If including case-control studies, the definition of the case must be included.

As this manuscript focuses on choline intake, the reader could understand that all the validated FFQs mentioned are validated for choline intake, which is incorrect and misleading. If using validated the nutrient or energy should be mentioned. 

The length or duration of the intake information collection must be included in all the studies included. How many days for dietary records or months for FFQs.

The time period should be included in weeks or months postpartum for lactation.

Discussion

Duplication of information in the text. For example, lines 152 and 166.

The definition of AI should be provided. The interpretation in line is inaccurate and needs to be revised. It should also be mentioned that a requirement for choline has not been set.

Make sure that the statement in line 182 is correct. How will you increase the phospholipid content in foods after cooking?

Specify if the study mentioned in line 190 was conducted in humans or not.

It is not clear what the authors refer to as negative coefficient in line 189.

The study mentioned in line 200 used serum, please keep in mind the limitations of using serum instead of plasma to quantify free choline concentrations. 

Recent relevant papers should be included and discussed in relation to different diets in line 201.

Concentrations of Water-Soluble Forms of Choline in Human Milk from Lactating Women in Canada and Cambodia.
Wiedeman AM, Whitfield KC, March KM, Chen NN, Kroeun H, Sokhoing L, Sophonneary P, Dyer RA, Xu Z, Kitts DD, Green TJ, Innis SM, Barr SI.
Nutrients. 2018 Mar 20;10(3):381. doi: 10.3390/nu10030381.
PMID: 29558412 

Total Water-Soluble Choline Concentration Does Not Differ in Milk from Vegan, Vegetarian, and Nonvegetarian Lactating Women.
Perrin MT, Pawlak R, Allen LH, Hampel D.
J Nutr. 2020 Mar 1;150(3):512-517. doi: 10.1093/jn/nxz257.
PMID: 32133524 

As this manuscript is about choline, line 205 contradicts what was described before, when it was mentioned that choline intake did not significantly change.

In the introduction and discussion, it is mentioned that a plant-based diet could decrease choline intake. However, line 209 now talks about F&V but no link to choline is included.

The topic of choline supplements should be discussed in greater detail. How much? what is the gap? which form? as part of MVM or by itself?

Line 214 the choline UL should not be interpreted as within, who set the UL?. It is necessary to mention requirements (EAR or AR) before referring to RDA.

Please make sure that the final evaluation and conclusions (as the abstract) match what has been presented in the body of the review. As relevant aspects are not discussed at this moment.

Author Response

(The authors gave the same response as above.)

Round 2

Reviewer 2 Report

The authors have considerably improved the manuscript.

I have only 1 small comment and 1 suggestion.

Comment

In line 15, it looks like the word 'high' is missing in the text.

Also, it is not clear to me the rationale for excluding studies based on choline intakes >400 mg/d. Keep in mind that the recommended AIs (EFSA and IOM) for this group range from 400 to 550 mg/d. So, why exclude those studies?

The rationale for excluding intakes <200 mg/d is also missing.

In addition, this relevant information should be included in the methods section. 

Suggestion

2) consider using the word 'prenatal' instead of ''antenatal'' when referring to supplements.

Author Response

In line 15, it looks like the word 'high' is missing in the text.

Thank you this has been added.

Also, it is not clear to me the rationale for excluding studies based on choline intakes >400 mg/d. Keep in mind that the recommended AIs (EFSA and IOM) for this group range from 400 to 550 mg/d. So, why exclude those studies?

The rationale for excluding intakes <200 mg/d is also missing.

In addition, this relevant information should be included in the methods section.

Thank you, we excluded these as they were outliers possibly due to cultural and methodological issues.   This has been explained in further detail in the methods (extra information has been added; highlighted in yellow).  We have also now also excluded the Carmichael study as the data was presented in percentiles, so even though this appeared high it was not a mean value, so we thought best to remove it per se.

Suggestion

2) consider using the word 'prenatal' instead of ''antenatal'' when referring to supplements.

Thank you this has been changed throughout (highlighted in yellow).
